# Traumatic events and psychological wellbeing among Palestinians: the moderating roles of mattering, anti-mattering and posttraumatic growth

traumatic events; mattering; posttraumatic growth; psychological well-being; Palestine

**Corresponding author:**
Dana Bdier;
Email: d.bdair@najah.edu

Dana Bdier 🄳, Fayez Mahamid 🄳 and Bilal Hamamra

Faculty of Humanities and Educational Sciences, An-Najah National University, Nablus, Palestine

## Abstract

The current study aimed to test the association between traumatic events and psycholgocial wellbeing among Palestinians, and to explore whether mattering, anti-ant-mattering, and posttraumatic growth (PTG) moderate the association between these two variables. A total of 610 Palestinian adults participated in the study, comprising 220 males and 390 females. Participants were recruited using online methods, including emails, social media, and advertisements. Results of correlational analysis revealed that traumatic events showed a negative correlation with PTG ($r = -.19$, $p < .01$), psychological well-being ($r = -.22$, $p < .01$), and mattering ($r = -.17$, $p < .01$). In contrast, traumatic events were positively associated with anti-mattering ($r = .18$, $p < .01$). Results of regression analysis showed that psychological well-being was negatively predicted by traumatic events and anti-mattering, while it was positively predicted by mattering and PTG. The current study emphasizes the importance of creating interventions that promote PTG, allowing Palestinians to transform their traumatic experiences into opportunities for personal and communal development. Additionally, enhancing self-mattering is strongly associated with psychological well-being and resilience. Encouraging Palestinians to feel valued and connected to their communities may help mitigate the negative effects of prolonged trauma, ultimately leading to better mental health outcomes.

## Impact statement

The results of this study provide important insights into the relationship between trauma, well-being, and the mediating roles of mattering, anti-mattering, and posttraumatic growth (PTG) among Palestinians. This research demonstrates that the importance of promoting PTG and self-mattering is further supported by the identification of key predictors of psychological well-being. Through capacity building, these strategies explain how Palestinians can turn the trauma they live through into sources of growth for themselves and their community to help them develop resilience and enhance mental health in the face of long-term political trauma. The research provides a basis for creating targeted interventions to improve well-being and resilience in coping with trauma.

## Theoretical background

Since 1948, Palestinians have endured catastrophic events, beginning with the forced displacement of over 700,000 individuals who became refugees scattered in neighboring countries (Mahamid, 2020). Over the course of 77 years, the Israeli military occupation has subjected Palestinians to a series of traumatic events, including military incursions, imprisonment, land confiscations, home demolitions and evictions, arrests, killings and torture (Mahamid et al., 2024).

Palestinians have experienced various traumatic events, both directly and indirectly, including witnessing mutilated bodies on television, exposure to intense artillery shelling, observing the aftermath of bombardments, hearing the deafening sounds of jet fighters, airstrikes, drones, witnessing the killing of loved ones and being used as human shields by the military soldiers (Thabet et al., 2014, 2018). Such exposure to violence which has been ongoing against Palestinians for 77 years has significantly impacted their psychological well-being. Indeed, many research papers demonstrated that experiencing war-related traumas increases the risk of psychological distress, particularly when accompanied by the loss of a family member (El-Khodary et al., 2020; Mahamid et al., 2023a).

In broader terms, well-being refers to the degree to which an individual feels satisfied, experiences happiness, and finds enjoyment in life (Mahamid and Bdier, 2020). Psychological



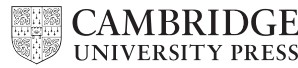

well-being, on the other hand, is more comprehensively described as a process of self-fulfillment, encompassing six key dimensions: autonomy, mastery over one's environment, personal development, meaningful relationships with others, a sense of purpose in life and self-acceptance (Weiss et al., 2016).

In a study that examined the relationship between exposure to political traumatic events and psychological well-being among Palestinians, the results of the study showed a negative relationship between exposure to the political traumatic events and psychological well-being among the study sample (Mahamid et al., 2021). Moreover, Mahamid (2022) explored the correlation between war-related quality of life, depression and hopelessness among Palestinians. The results revealed a negative correlation between the study variables. Furthermore, Mahamid et al. (2023b) found that political violence positively associated with posttraumatic stress symptoms among Palestinian adults.

## Mattering and anti-mattering

Regarding this study, the experience of mattering is expected to be a protective factor against traumatic events, as it emphasizes the essential connection between individual and social dimensions, which is crucial for achieving a holistic sense of well-being (Paradisi et al., 2024). Mattering, was originally defined by Rosenberg and McCullough (1981) as the feeling of being important and significant to others. According to James (1890), even individuals with a strong sense of self-worth, such as high self-esteem, cannot attain complete well-being if they are not valued or acknowledged by others. In other words, well-being is based on intersubjective relations. This makes mattering particularly relevant to well-being, as it links the personal and relational aspects of human experience.

Rosenberg and McCullough (1981) outlined three key components of mattering: the belief that others rely on us, the perception that we are valued by others, and the awareness that others pay attention to us. Later, Rosenberg (1985) introduced a fourth element: the sense that we would be missed if we were no longer present.

Mattering plays a crucial role in promoting mental well-being, as it is linked to lower levels of depression, suicidal thoughts and other negative health outcomes (Elliott et al., 2004; Milner et al., 2016; Taylor and Turner, 2001). Moreover, research demonstrated that mattering contributes to explaining variations in psychological outcomes, independently of other related constructs such as self-esteem, social support and mastery (Flett et al., 2022). Furthermore, individuals with a strong sense of mattering are more likely to possess greater psychological resources, which help them traverse challenges in their environment, which enables them to resist mental illness and maintain confidence in achieving future goals (Liu et al., 2023).

On the other hand, anti-mattering refers to the feeling of not being valued or significant to others (Nepon and Flett, 2024). Unlike simply being the opposite of mattering, anti-mattering involves a distinct emotional and motivational experience. It arises from negative automatic thoughts such as "I don't matter," "I'm not important," or "I'm invisible," which reflect a deep sense of being overlooked or disregarded by others (Nepon and Flett, 2024).

Anti- mattering is expected to be a risk factor against the psychological well-being of Palestinians, as it conveys an intensely painful feeling of being both irrelevant and isolated, where the individuals not only feel unnoticed and alone but also believe that their presence and voice are unheard by others (Flett, 2024). In addition, anti-mattering was found to be positively correlated with insecure attachment, self-criticism, dependency and low levels of well-being (Flett et al., 2022).

## Posttraumatic growth

Posttraumatic growth (PTG) refers to the positive transformation individuals may undergo following significant life crises. It arises as a result of grappling with profound challenges and tends to manifest more clearly in the context of severe stressors rather than everyday stressors. Unlike mere coping mechanisms, PTG is perceived as a deeper, outcome-oriented process that fosters genuine personal growth. This growth is often accompanied by profound and enduring life changes, going beyond superficial illusions of improvement. Distinct from concepts like thriving or flourishing, PTG frequently necessitates a fundamental reevaluation of core beliefs and assumptions about one's life and existence (Tedeschi et al., 2015).

In the current study, PTG is predicted to be a protective factor against the negative effects of exposure to traumatic events on mental health among Palestinians. This is similar to Veronese et al. (2017) study which examined the relationship between PTG and subjective wellbeing among Palestinian professional helpers from the Gaza Strip and West Bank after being exposed to traumatic events. The results showed that posttraumatic growth was found to have a positive direct relationship with subjective well-being among the study sample. Furthermore, Dawwas and Thabet (2017), explored the association between traumatic experience, posttraumatic stress disorder, resilience, and posttraumatic growth among youth in Gaza Strip. The results of the study revealed a positive relationship between total traumatic events due to war and PTSD and negative correlation between PTG and resilience. While PTSD was negatively correlated with resilience, PTG was positively correlated with resilience.

## Current study

Based on previous findings (Dawwas and Thabet, 2017; Elliott et al., 2004; Flett, 2024; Flett et al., 2022; Liu et al., 2023; Mahamid et al., 2021, 2022, 2023b; Milner et al., 2016; Taylor and Turner, 2001; Veronese et al., 2017), four study hypotheses were defined: (1) traumatic events will be negatively associated with psychological wellbeing among Palestinian adults; (2) mattering will mediate the association between traumatic events and psychological well-being among Palestinian adults; (3) anti-mattering will mediate the association between traumatic events and psychological well-being among Palestinian adults; (4) posttraumatic growth will mediate the association between traumatic events and psychological well-being among Palestinian adults.

## Methodology

### Participants and procedures

The research was conducted in April 2024 and targeted Palestinian youth living in the West Bank and Gaza Strip. Participants were recruited using online methods, including emails, social media, and advertisements. The aims of the study were presented online, and participants interested in participating were asked to send an email indicating their willingness to participate in the study. All participants received a letter clarifying the objectives and ethical issues of the study. They provided written informed consent upon accepting

the conditions of participation. A total of 610 Palestinian adults participated in the study, comprising 220 males and 390 females. Regarding educational attainment, 34.73% held a graduate degree, 55.12% held a bachelor's degree and 10.15% held a high school degree. To be included in the study, participants were required to be (1) native Arabic speakers, (2) Palestinian, and (3) residents in the occupied Palestinian territories (oPt). Approval for the study was obtained from An-Najah National University IRB before data collection was initiated.

## Measures

Following standard methodological recommendations for questionnaire development (Hambleton and Lee, 2013), all measures not already validated in Arabic were translated and back-translated from the original English version into Arabic. This process involved a panel of 10 Arab professionals in psychology, counseling and social work who evaluated the clarity and relevance of the questions and translations. After completing the initial draft of translated items, the questionnaires were back-translated into English by an independent expert English editor. Based on the editor's feedback, the translated version was pilot-tested among 80 participants and further refined for clarity.

*Brief Trauma Questionnaire (BTQ)*: The BTQ (Schnurr et al., 1999) is a 10 item self-report measure used to determine if responders have ever experienced a traumatic event. This measure assesses inquiry about events, such as experiencing combat, a motor vehicle accident or the sudden death of a close friend or family member. In either case, exposure to an event should be scored as positive if a respondent says yes to either. Reliability analysis of BTQ indicated a high degree of reliability in evaluating traumatic events of Palestinians ($\alpha = .88$).

*Posttraumatic Growth Inventory-Short Form (PTGI-SF).* The PTGI-SF) is a 10 items self-report with response options from 1 (entirely in disagreement) to 6 (entirely in agreement), (Cann et al., 2010). The PTG dimensions (new possibilities, relating to others, personal strength, appreciation of life and spiritual change). We used an Arabic version of the instrument, validated in a Palestinian sample of in-service health providers (Veronese and Pepe, 2019). In the present study, Chronbach's α formula indicated a high internal consistency of PTGI-SF in the Palestinian context (.89).

*The Warwick-Edinburgh Mental Well-being Scale (WEMWBS)*: The WEMWBS is 14 items covering both hedonic and eudaimonic aspects of mental health, including positive affect (feelings of optimism, cheerfulness and relaxation), satisfying interpersonal relationships and positive functioning (energy, clear thinking, self-acceptance, personal development competence and autonomy). Individuals completing the scale must tick the box that best describes their experience of each statement over the past two weeks using a 5-point Likert scale (none of the time, rarely, sometimes, often, all of the time). The Likert scale represents a score for each item from 1 to 5 (Tennant et al., 2007). An example of scale items is "I've been feeling optimistic about the future." The WEMWBS indicated a high internal consistency in assessing Palestinians' global health ($\alpha = .88$).

*The General Mattering Scale (GMS).* The GMS (Marcus and Rosenberg, 1987) has five items measuring perceptions and feelings of being important to other people (e.g., "How much would you be missed if you went away?"). Items are rated on a scale ranging from 1 (Not at all) to 4 (A lot). Psychometric analyses have shown that this scale is unidimensional, with good reliability and validity (Taylor and Turner, 2001). Higher scores reflect greater levels of mattering (Marcus and Rosenberg, 1987).

*Anti-Mattering Scale (AMS).* The AMS is a five-item unifactorial scale assessing the degree to which people feel like they do not matter to others. A representative item is, "How often have you been treated in a way that makes you feel like you are insignificant?" Items are rated from 1 (not at all) to 4 (a lot). Higher scores reflect greater levels of anti-mattering. The instructions are: choose the rating you feel is best for you based on your experiences with people in general. For each item, please circle a number to indicate your response (Flett et al, 2022).

## Data analysis

We used descriptive statistics, means, standard deviations, range, skewness, kurtosis and reliability for our study variables (traumatic events, mattering, posttraumatic growth, anti-mattering and psychological well-being). In addition, Person correlation coefficient among traumatic events, mattering, posttraumatic growth, anti-mattering and psychological well-being was conducted to evaluate whether there is statistical evidence for a linear relationship among our study variables. Finally, we conducted a hierarchical regression analysis to examine the predictors of psychological well-being. In Step 1, demographic variables (gender and educational level) were included. In Step 2, demographic variables (gender and educational level) along with traumatic events were used to predict psychological well-being. Finally, Step 3 included demographic variables (gender and educational level), traumatic events, mattering, anti-mattering and posttraumatic growth to predict psychological well-being. The hierarchical regression analysis has been tested using SPSS 29 software for data analysis.

## Findings

Descriptive statistics for mattering, anti-mattering, traumatic events, posttraumatic growth and psychological well-being are presented in Table 1. Participants showed high levels in mattering, traumatic events, posttraumatic growth and psychological well-being, while their scores in anti-mattering were average. Furthermore, all measures used in this study demonstrated a high degree of reliability, with coefficients ranging from 0.90 for posttraumatic growth to 0.94 for anti-mattering.

The correlational analysis results, shown in Table 2, revealed several significant associations. Mattering had a negative correlation with anti-mattering ($r = -.40$, $p < .01$) and traumatic events ($r = -.17$, $p < .01$), and with a positive correlation with posttraumatic growth ($r = .63$, $p < .01$) and psychological well-being ($r = .57$, $p < .01$). Moreover, anti-mattering was positively correlated with traumatic events ($r = .18$, $p < .01$) and negatively correlated with both posttraumatic growth ($r = -.40$, $p < .01$) and psychological well-being ($r = -.20$, $p < .01$). Traumatic events showed a negative correlation with posttraumatic growth ($r = -.19$, $p < .01$) and psychological well-being ($r = -.22$, $p < .01$). Lastly, posttraumatic growth was positively correlated with psychological well-being ($r = .54$, $p < .01$).

In Table 3, we conducted a hierarchical regression analysis to examine the predictors of psychological well-being. In Step 1, demographic variables (gender and educational level) were included. In Step 2, demographic variables (gender and educational level) along with traumatic events were used to predict psychological well-being. Finally, Step 3 included demographic variables (gender and educational level), traumatic events, mattering, anti-mattering, and posttraumatic growth to predict psychological well-being.

**Table 1.** Descriptive statistics for research variables (N = 610)

| Variable | M | S.D | Min | Max | Range | Skewness | Kurtosis | Cronbach's alpha |
|----------|---|-----|-----|-----|-------|----------|----------|------------------|
| Mattering | 3.42 | .56 | 1.20 | 4.00 | 2.80 | −.47 | .84 | .93 |
| Anti-mattering | 2.12 | .74 | 1.00 | 4.00 | 3.00 | .60 | −.08 | .94 |
| Traumatic events | 1.55 | .25 | 1.00 | 2.00 | 1.00 | .07 | −.82 | .91 |
| Posttraumatic growth | 3.99 | .60 | 2.30 | 5.00 | 2.70 | −.82 | .20 | .90 |
| Psychological well-being | 3.82 | .71 | 1.00 | 5.00 | 4.00 | −.44 | .67 | .92 |

**Table 2.** Correlations among study variables (N = 610)

| Measures | 1 | 2 | 3 | 4 | 5 |
|----------|---|---|---|---|---|
| Mattering | 1 | −.40[a] | −.17[a] | .63[a] | .57[a] |
| Anti-mattering | | 1 | .18[a] | −.40[a] | −.20[a] |
| Traumatic events | | | 1 | −.19[a] | −.22[a] |
| Posttraumatic growth | | | | 1 | .54[a] |
| Psychological well-being | | | | | 1 |

[a] $\alpha$ is significant at ≤ .01.

**Table 3.** Hierarchical regression analysis for variables predicting psychological well-being

| Variable | B | SEB | $\beta$ | R2 |
|----------|---|-----|---------|-----|
| *Step1* | | *Psychological well-being* | | .05 |
| *Educational level* | −.18 | .09 | −.08[a] | |
| *Gender* | −.16 | .04 | −.15[b] | |
| *Step 2* | | | | .16 |
| *Educational level* | −.19 | .09 | −.08[a] | |
| *Gender* | −.17 | .04 | −.15[b] | |
| *Traumatic events* | −.25 | .11 | −.16[b] | |
| *Step3* | | | | |
| *Educational level* | −.29 | .07 | −.10[a] | .40 |
| *Gender* | .04 | .037 | .14[b] | |
| *Traumatic events* | −.064 | .09 | −.17[b] | |
| *Mattering* | .558 | .05 | .44[b] | |
| *Anti-mattering* | −.087 | .03 | −.10[a] | |
| *Posttraumatic growth* | .376 | .04 | .31[b] | |

[a] $\alpha$ is significant at ≤ .05.
[b] $\alpha$ is significant at ≤ .01.

Our results showed that psychological well-being was negatively predicted by gender ($\beta = -.15$; ** $p < .01$), favoring male respondents (females, M = 3.71; males, M = 3.84), and by educational level ($\beta = -.08$; ** $p < .01$), favoring those with graduate studies (graduate studies, M = 3.90; BA, M = 3.88; high school diploma, M = 3.60). Additionally, psychological well-being was negatively predicted by traumatic events ($\beta = -.16$; ** $p < .01$) and anti-mattering ($\beta = -.10$; ** $p < .01$). On the other hand, psychological well-being was positively predicted by mattering ($\beta = .44$; ** $p < .01$) and posttraumatic growth ($\beta = .31$; ** $p < .01$).

## Discussion

The main objectives of the current study were to examine the relationship between traumatic events and psychological well-being, and to determine whether traumatic events, posttraumatic growth, mattering and anti-mattering could predict psychological well-being among Palestinians. The results of the study revealed a negative association between traumatic events and psychological well-being. In addition to this, the findings showed positive associations between mattering, posttraumatic growth and psychological well-being. Specifically, psychological well-being was negatively predicted by traumatic events and anti-mattering, while it was positively predicted by mattering and posttraumatic growth.

The findings of our study are in line with previous research that indicated psychological well-being is negatively predicted by traumatic events. For instance, Mahamid et al. (2024) examined whether traumatic events predict psychological well-being and the desire to migrate among Palestinians. Their findings revealed that psychological well-being and the desire to migrate were positively predicted by traumatic events. Uygun (2020) investigated the relationship between types of traumatic events, post-traumatic stress, and mental well-being among Syrians. The findings revealed that the most significant predictor of psychological well-being was the level of traumatic events related to war trauma. Mahamid et al. (2021) examined the relationship between stressful life events and psychological well-being among Palestinian adolescents. The findings indicated that psychological well-being was negatively and significantly predicted by traumatic events.

The findings of our study, which highlight the negative impact of traumatic events on psychological well-being among Palestinians, can be understood in the context of the traumatic experiences that have shaped the lives of Palestinians who have been subject to the brutal practices of the Israeli military occupation for decades. These events are largely rooted in the actions of the Israeli occupation, including the destruction of homes, military incursions into Palestinian cities, villages and refugee camps, and the establishment of a number of military checkpoints across the cities and villages of the West Bank. These checkpoints impede movement among the Palestinian cities and villages, putting the lives of Palestinians at risk.

The traumatic experiences faced by the Palestinian community have only intensified following the events of October 7, 2024. Since then, the frequency of military raids on Palestinian cities and villages, as well as refugee camps, has drastically increased, resulting in heightened violence and daily aggression and raids against Palestinian cities, camps and villages. In addition, the treatment of Palestinians at military checkpoints has worsened, with more instances of harassment, humiliation and physical abuse. The military occupation has escalated its use of violence, implementing more severe 'preventive measures' with increasing regularity.

These developments have exacerbated the psychological distress already prevalent among Palestinians. Moreover, Palestinianshave been enduring a deep-rooted, transgenerational trauma since the Palestinian Nakba of 1948. The majority of Palestinians are refugees who have lived in harsh, often inhumane conditions for decades. Their lives are marked by extreme poverty, displacement, and a constant sense of insecurity, preventing them from achieving even the basic level of psychological well-being and human rights. This ongoing Nakba and the continuous exposure to traumatic events have a lasting influence on the Palestinians" mental health and well-being, an influence that is transgenerational.

The findings of our study indicated that mattering and posttraumatic growth play a moderating role in mitigating the effects of traumatic events and improving psychological well-being among Palestinians. This is consistent with previous research that highlights the mediating and moderating roles of these factors in the relationship between traumatic events and psychological well-being. Veronese et al. (2022) found that posttraumatic growth positively mediated the relationship between stressful life events and psychological well-being among Palestinians during the COVID-19 pandemic. Similarly, Giangrasso et al. (2022) found that mattering and self-esteem were unique predictors of psychological well-being levels among Italians during the same period. Their results suggested that individuals who feel they do not matter may be particularly vulnerable to stress, depression and anxiety, which can lead to a decline in psychological well-being.

Based on this, both the variables of feeling mattering and posttraumatic growth can be considered important mediating factors that help individuals overcome the traumatic experiences they have gone through. Feeling mattering represents a sense of social acceptance by others, whereas Palestinians perceive that they receive attention and emotional care from others. This can effectively contribute to helping individuals overcome various hardships and traumatic events, reducing the deterioration of their mental health. Similarly, posttraumatic growth, which individuals develop after traumatic events, represents a form of self-awareness of those experiences, acceptance of loss, and the development of a positive life path. These two factors work together to alleviate feelings of pain and loss and contribute to reducing the impact of traumatic events on Palestinians.

Regarding the demographic variables in the study, the results showed that males reported higher levels of psychological well-being compared to females in the Palestinian community. This may be attributed to the cultural and social context within Palestinian society, where males tend to have more opportunities in both social and professional spheres than females. This advantage could lead to greater psychological well-being for males compared to females. Furthermore, our findings revealed that individuals with higher academic qualifications, such as a master's or doctoral degree, reported higher levels of psychological well-being, which may help mitigate the impact of stressors brought about by the Israeli occupation. This could be related to the better employment opportunities available to educated individuals, making them less likely to experience the negative effects of traumatic events related to the political situation in Palestinian society. The working class in Palestine, particularly those without academic degrees, has faced significant unemployment, especially after the disruption of the job market in Israel, which has further intensified the hardships faced by the Palestinian working class. Furthermore, education has been a key factor in enabling Palestinians to maintain resilience in the face of the ongoing Israeli occupation of Palestinian territories. It can also help in the development of effective coping strategies among Palestinians in general.

## Limitations

Despite these contributions, some limitations must be acknowledged. The cross-sectional design of the study restricts the ability to draw causal conclusions and limits the generalizability of the results. While the associations observed are promising, future experimental and longitudinal studies are needed for further validation. The online recruitment method may have excluded affected and clinical populations within the Palestinian context. Future research should aim to include these groups for a more comprehensive understanding of how mattering, anti-mattering and posttraumatic growth mediate the association between traumatic events and psychological well-being. Moreover, using traditional pencil-and-paper questionnaires could provide valuable insights, particularly for affected groups. The study's sample included only participants from the West Bank of Palestine. Future studies should target multiple regions of Palestine, such as East Jerusalem and the Gaza Strip, to enhance the generalizability of the findings.

## Conclusion

The current study examined the association between traumatic events and psychological well-being among Palestinians, as well as to test whether mattering, anti-mattering and posttraumatic growth serve as moderators in this relationship. The findings revealed that traumatic events negatively predicted psychological well-being. On the other hand, posttraumatic growth and anti-mattering positively predicted psychological well-being, suggesting that these factors serve as effective moderating variables between traumatic events and psychological well-being.

The results of the current study emphasize the need to design therapeutic interventions that are not only tailored to the feverish Palestinian context but also take into account the cultural and socio-political challenges faced by Palestinian. Such interventions should recognize the impact of prolonged exposure to trauma, including the effects of the Israeli Zionist occupation with its consequence of displacing Palestinians and demolishing their houses and arresting, torturing and killing them, practices that shape the Palestinians' collective identity and trauma and their psychological wellbeing.

These interventions should focus on promoting posttraumatic growth, enabling Palestinians to transform their traumatic experiences into opportunities or subjective spaces of personal and communal development. By fostering growth after trauma, these interventions can help Palestinians regain a sense of agency and empowerment despite the Israeli ongoing violence against them. Furthermore, boosting self-mattering is essential, as it is closely linked to psychological well-being and resilience. Encouraging Palestinians to feel valued and connected to their communities may reduce the negative effects of trauma, leading to improved mental health outcomes.

Moreover, strengthening mental health professionals' competencies in Palestine is crucial for the effectiveness of these interventions. Given the complexity of trauma and the ongoing colonial violence, it is essential that therapists and counselors are equipped with the necessary skills and knowledge to address the specific

needs of the Palestinian population. This includes understanding the cultural dynamics, historical trauma and social stressors that Palestinians experience daily.

In sum, developing contextually appropriate therapeutic approaches and enhancing the capabilities of mental health professionals in Palestine is vital for improving psychological well-being, particularly for those most directly affected by traumatic experiences. Such efforts can contribute to the overall resilience of the Palestinian community, offering hope and support as they traverse the enduring challenges they face.

**Open peer review.** To view the open peer review materials for this article, please visit http://doi.org/10.1017/gmh.2025.34.

**Data availability statement.** The datasets used and/or analyzed during the current study available from the corresponding author on reasonable request.

**Author contributions.** All authors contributed equally to this article.

**Competing interest.** The authors declare that they have no conflict of interest. All authors agreed in submitting the manuscript to the journal.

**Ethics approval and consent to participate.** All procedures performed in this study involving human participants were in accordance with the ethical standards of An-Najah National University IRB, the American Psychological Association, and with the Helsinki Declaration. Informed consent was obtained from parents all participants. The protocol of our study was received ethical approval from An-Najah National University IRB before data collection was initiated.

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
