## [Reviewer Report]

Much thanks for your contribution to this much needed area.

One concern about the paper is the use of Post-Traumatic Growth as a construct. In Palestine there has not yet been a period of post trauma as the incidents of violence have never ended and have actually increased significantly over the time of the study. This variable should be removed or explained as it does not seem a true analysis in the current Palestinian context.

Secondly the highly politicized language should be removed as it reduces the credibility of the article. While it is clear that the aggression of the Israeli occupying forces are the perpetrators of violence in Palestine (both in the West Bank and Gaza) there are multiple factors that negatively affect the lives of Palestinians ( including violence from teh PA, poverty, etc.). Reducing the political rhetoric in the narrative will create an environment where the data from the study can be examined in a neutral scientific space.

---

## [Reviewer Report]

Thank you for the article. Please review the references within text and at the end and ensure that all according to APA.

---

## [Editor Report]

Dear Dr. Bdier,

I hope this message finds you well. I am pleased to inform you that after careful consideration, your manuscript titled “Traumatic Events and Psychological Wellbeing Among Palestinians: The Moderating Roles of Mattering, Anti-Mattering, and Posttraumatic Growth” has been accepted for publication in Cambridge Prism: Global Mental Health.

We believe your research makes a significant contribution to the field and will be valuable to our readership. Our editorial team will now begin the process of preparing your manuscript for publication, and you will be contacted with further details regarding the next steps.

Thank you for submitting your important work to Cambridge Prism: Global Mental Health. We look forward to sharing your research with the wider academic community.

If you have any questions or need further assistance, please feel free to reach out.

Best regards,

Sara Romero